# Effects of Different Bedside Physiotherapy Frequencies in Hospitalized COVID-19 Patients, Focusing on Mild to Moderate Cases

**DOI:** 10.3390/ijerph22060931

**Published:** 2025-06-12

**Authors:** Netchanok Jianramas, Thanaporn Semphuet, Veeranoot Nissapatorn, Chaisith Sivakorn, Maria de Lourdes Pereira, Anuttra (Chaovavanich) Ratnarathon, Chenpak Salesingh, Eittipad Jaiyen, Salinee Chaiyakul, Nitita Piya-Amornphan, Thanrada Thiangtham, Kornchanok Boontam, Khomkrip Longlalerng

**Affiliations:** 1Department of Physical Therapy, School of Allied Health Sciences, Walailak University, Nakhon Si Thammarat 80160, Thailand; netchanok.j@tsu.ac.th (N.J.); thanaporn.s@tsu.ac.th (T.S.); csalinee@wu.ac.th (S.C.); nitita.do@wu.ac.th (N.P.-A.); thanradagungun@gmail.com (T.T.); kookkiksmily@gmail.com (K.B.); 2Movement Science and Exercise Research Center-Walailak University (MoveSE-WU), Nakhon Si Thammarat 80160, Thailand; 3Department of Physical Therapy, Faculty of Allied Health Sciences, Thaksin University, Phatthalung 93210, Thailand; 4Department of Medical Technology, School of Allied Health Sciences, Walailak University, Nakhon Si Thammarat 80160, Thailand; nissapat@gmail.com; 5Futuristic Science Research Center, School of Science, Walailak University, Nakhon Si Thammarat 80160, Thailand; 6Intensive Care Unit, University College London Hospitals, London NW1 2BU, UK; chaisith@live.com; 7Intensive Care Unit, Bangkok Hospital, Bangkok 10310, Thailand; 8CICECO-Aveiro Institute of Materials, Department of Medical Sciences, University of Aveiro, 3810-193 Aveiro, Portugal; mlourdespereira@ua.pt; 9Department of Internal Medicine, Bamrasnaradura Infectious Disease Institute, Nonthaburi 11000, Thailand; parnmed12@hotmail.com; 10Physical Therapy Department, Bamrasnaradura Infectious Disease Institute, Nonthaburi 11000, Thailand; chuenpak.s@gmail.com (C.S.); eittipad.jaiyen@gmail.com (E.J.)

**Keywords:** physiotherapy, rehabilitation, SARS-CoV-2

## Abstract

Currently, knowledge of the effects of different frequencies of administration of bedside physiotherapy programs (PTPs) on hospitalized COVID-19 patients is limited. Therefore, this study aimed to compare the effects of administering PTPs once or twice during hospitalization versus daily PTPs until discharge. Fifty-two COVID-19 patients were equally assigned to two groups, matched by gender and age (1:1 ratio). Experimental Group 1 (Ex-G1) received PTPs one to two times during hospitalization, while Experimental Group 2 (Ex-G2) received daily PTPs until discharge. The outcomes assessed included the survival rate, length of hospitalization (LoH), intensive care unit (ICU) referrals, and in-hospital complications. Most participants were classified as having mild to moderate COVID-19, with a mean age of 45 years. No significant differences were observed between the groups in all primary outcomes, including the survival rate (*p* = 1.000), LoH (*p* = 0.117), ICU referrals (*p* = 0.313), and complications (*p* = 0.555). The overall survival rate was 98%. One Ex-G2 participant was referred to the ICU, while complications occurred in two Ex-G1 and four Ex-G2 participants. In summary, for patients with mild to moderate COVID-19, one to two bedside physiotherapy sessions produced comparable results to daily physiotherapy in terms of the survival rate, LoH, ICU referrals, and in-hospital complications.

## 1. Introduction

Since 2019, human coronavirus disease 2019 (COVID-19) has spread throughout the world, including Thailand [1,2]. This emerging disease directly affects the patient’s respiratory system [3]. Its clinical signs and symptoms include fever; cough; chills; short, shallow, and difficult breathing; fatigue; malaise; headache; anosmia; ageusia; sore throat; stuffy nose and runny nose; nausea and vomiting; and diarrhea [3,4,5]. Patients with more severe conditions experience more pronounced signs and symptoms, which are caused by pneumonia [6]. Physiotherapy programs (PTPs) are commonly suggested for adults with pneumonia who are intubated and mechanically ventilated, promoting the clearance of secretions and lung compliance [7]. Experienced physicians, researchers, and physiotherapists from various countries have suggested that PTPs should be applied to COVID-19 patients if there is evidence of pneumonia without exudate consolidation, mucous hypersecretion and difficulty clearing secretions, functional decline, and (a risk of) intensive care unit (ICU)-acquired weakness [8,9,10]. PTPs include prone positioning, postural drainage, breathing exercises and devices (e.g., positive expiratory pressure and inspiratory muscle training), and ventilator use, as well as functional training, exercise, and early mobilization [9,11,12,13].

Notably, most experts’ suggestions and some studies agree that patients with severe symptoms or those discharged from the intensive care unit derive benefits from PTPs [9,11,12,13,14]. Conversely, some researchers have suggested that PTPs are contraindicated during the acute phase, since the acute effects of the exercise program may cause an increase in pro-inflammatory cytokines and viral replication [15,16]. Nevertheless, a recent study has shown an improvement in immune function after two weeks of moderate aerobic exercise [17], which is supported by previous review studies [18,19]. Interestingly, most previous studies have investigated the effects of PTPs in the acute or sub-acute phase in severely to critically ill patients with COVID-19 [11,12,14,20,21]. However, there is a lack of research involving patients with mild to moderate COVID-19, despite this group representing most cases globally. A narrative review suggests that conventional PTPs and physical exercise may be appropriate for patients with mild to moderate disease severity [13]. Nevertheless, the evidence regarding the benefits of PTPs during the acute phase in this population remains inconclusive [9].

To our knowledge, bedside PTPs should be initiated as early and as frequently as possible in hospitalized patients. One study found that increasing the daily frequency of bedside physiotherapy sessions was most effective in reducing mortality rates, the length of hospitalization (LoH), and the incidence of respiratory infections among ICU patients [22]. In the context of COVID-19, only one study has demonstrated a positive association between the frequency of physical therapy visits and improvements in patients’ mobility status [23]. However, knowledge of the effects of varying bedside PTP frequencies in COVID-19 patients with primarily mild to moderate symptoms remains limited. Thus, this study aimed to compare the effects of different administration frequencies of bedside PTPs on the survival rate, LoH, referrals to the ICU, and in-hospital complications. In addition, the safety of patients during and after performing PTPs was investigated. We hypothesized that COVID-19 patients receiving daily bedside PTPs would have a significantly higher survival rate and lower LoH, as well as fewer complications, compared to those receiving less frequent bedside PTPs. In addition, none of the COVID-19 patients had serious adverse events during and after PTPs, and no physiotherapists tested positive for COVID-19 infection.

## 2. Materials and Methods

### 2.1. Study Design

A prospective, quasi-experimental study design was used to determine the effects of PTPs in the acute phase in patients with COVID-19. This study was conducted at the Bamrasnaradura Infectious Disease Institute, Nonthaburi Province, Thailand, from November 2021 to March 2022. This study was registered with Thai Clinical Trials, https://www.thaiclinicaltrials.org/, TCTR20210823004.

### 2.2. Study Population

Participants were stratified by age and gender and then assigned to two groups sequentially, based on the order of their hospital admission. The use of a randomized control group was limited due to ethical considerations, as well as constraints related to time and budget. General inclusion criteria included adults aged 18–65 years old with (1) a nucleic acid test-confirmed diagnosis of SAR CoV-2 infection, (2) hospitalization due to any clinical signs and symptoms of pneumonia, (3) no communication problems, and (4) ability to use an online mobile phone application. Regarding the specific inclusion criteria, patients were recruited if one of the following criteria [8,9] was indicated: (1) COVID-19 with risk factors for severe disease, (2) confirmed case of pneumonia with hypoxia (resting blood oxygen saturation (SpO_2_) < 96% or the presence of exercise-induced hypoxemia defined as a reduction in SpO_2_ > 3% compared to baseline), (3) inability to expel secretions caused by prolonged immobilization and respiratory muscle weakness, (4) presence of breathlessness or dyspnea needing oxygen therapy (presence of signs of pneumonia with lung consolidation by chest radiograph or computerized tomography or lung ultrasound), and (5) presence of functional limitation caused by prolonged hospitalization, prolonged ICU stay, or prolonged use of a respirator or oxygen device. Exclusion criteria included (1) unwillingness or inability to follow the study protocol and (2) active participation in another study.

### 2.3. Characteristics and Classification of COVID-19 Disease Severity

COVID-19 patients were categorized as mild, moderate, severe, and critical according to the World Health Organization definition [24]. Mild severity was defined as having the following symptoms: fever, cough, fatigue, anorexia, shortness of breath, and myalgia. Moderate severity was identified in patients who were diagnosed as having pneumonia or hypoxia, with more pronounced signs and symptoms, including SpO_2_ > 90% in room air. Severe COVID-19 was defined as pneumonia accompanied by any signs and symptoms, including respiratory rate > 30 times/min and SpO_2_ < 90% in room air. Critical disease was defined as a patient with ARDS and in need of intensive oxygen therapy [24]. In addition, a laboratory investigation was used to determine the severity of patients’ conditions at baseline, including the open reading frame gene 1ab (ORF1ab) and the envelope (E) gene of SARS-CoV-2, a complete blood count (CBC), kidney and liver function, tissue damage markers (lactate dehydrogenase (LDH)), inflammatory markers (i.e., erythrocyte sedimentation rate (ESR) and C-reactive protein (CRP)), chest radiography, and medications.

### 2.4. Intervention, Physiotherapy Programs (PTPs)

Participants in Experimental Group One (Ex-G1) received only a one-time bedside PTP for the first few days after hospital admission. However, some patients remained confused about the program and could not perform it correctly; thus, a second bedside PTP was applied in these patients. Meanwhile, participants in Experimental Group Two (Ex-G2) had daily bedside PTPs until hospital discharge. Patients in both groups were taught about PTPs as soon as possible after being admitted to the hospital. PTPs comprised breathing exercises, secretion removal techniques (coughing, huffing, and positioning), active chest trunk mobilization, active exercise of both limbs, early progressive mobilization/ambulation, education, and psychosocial support [9,11,12,13] (Figure 1). The BreatheMax^®^V.2 device (C&D Biomedical, Bangkok, Thailand), which operates based on the principle of positive expiratory pressure (PEP) and an oscillating incentive spirometer (OIS), was given to Ex-G2 patients. All patients were invited to join a closed private group via a mobile application platform. Sosme were monitored and encouraged through direct communication via the app or ward phones, particularly participants in the Ex-G1 group, who received only a one-time PTP session. Short instructional video clips demonstrating the physiotherapy programs (PTPs) were shared within the secure group. Physiotherapists were available to address any questions or concerns, and they provided daily motivational messages to support adherence. Additionally, some participants voluntarily shared their exercise experiences, fostering a supportive environment that may have further encouraged adherence among others.

### 2.5. Safety Considerations for COVID-19 Patients in Assigning Physiotherapy Programs

Prior to performing the bedside PTPs, all patient data, vital signs, and medical records were intensively reviewed by physiotherapists. Patients were questioned about their current signs and symptoms. Permission was not granted for physiotherapist treatment if the patient had any of the following conditions, regardless of whether they were on a ventilator: fraction of inspired oxygen (FiO_2_) > 0.6, SpO_2_ < 90%, respiratory rate (RR) > 40 times/min, positive end-expiratory pressure (PEEP) > 10 cmH_2_O, ventilator resistance, unstable cardiovascular signs (systolic blood pressure > 180 mmHg or diastolic blood pressure > 110 mmHg, heart rate < 40 or > 120 beats/min) [25]. If SpO_2_ decreased > 3% from baseline during the PTP session, they were not allowed to continue the session.

### 2.6. Safety Considerations for Physiotherapists to Prevent SARS-CoV-2 Transmission

The physiotherapists were trained to use personal protective equipment (PPE) properly before performing the bedside PTPs [9]. Importantly, all physiotherapists had to strictly follow aerosol, airborne, and contact precautions [9] (Figure 1). Regarding the physiotherapists’ safety considerations, all of them were required to have (1) a minimum of two doses of the COVID-19 vaccine before the initiation of the study for at least 2 weeks, (2) good health with no comorbidities, (3) an age of <45 years [9], and (4) at least two years of experience in chest physiotherapy (Figure 1). All physiotherapists were tested weekly for COVID-19 infection using a reverse transcription polymerase chain reaction (RT-PCR) and rapid antigen test kit (ATK) as described in Section 2.8 (Laboratory Tests).

### 2.7. Outcome Measures

The primary outcomes were the survival rate, LoH, number of patients who were referred to the ICU, and rates/types of complications. The secondary outcome was the safety of patients, which was indicated by the minor and serious adverse events that occurred during and after each physical therapy session. A minor adverse event was defined as an event that slightly affected the patient, requiring time for rest and with recovery within 15 mins—for example, dizziness, nausea and vomiting, postural hypotension (blood pressure drop > 10 mmHg from baseline), fatigue, and a SpO_2_ drop greater than 3% from baseline. A serious adverse event was defined as one requiring urgent assistance—for example, medications or resuscitation. In addition, the other secondary outcome was the safety of the physiotherapists. The physiotherapists were tested for COVID-19 infection using an RT-PCR and ATK every week.

### 2.8. Laboratory Tests

Blood samples were collected from the antebrachial area in all participants. RT-PCR was performed using a Cobas^®^ 6800 SARS-CoV-2 assay on the Cobas^®^ 6800 platform (Roche Diagnostics, Basel, Switzerland) to identify the presence of the ORF1ab and E genes of SARS-CoV-2. The ATK was applied using a Singclean device (Hangzhou Singclean Medical Products, Hangzhou, China). The CBC was examined using an electrical impedance device (Mindray Model CAL 6000, Shenzhen, Guangdong Province, China). Meanwhile, renal function, liver function, and LDH were examined using photometry (Beckman Coulter, Brea, CA, USA). The ESR was examined using the Westergren method on a Mini-VES analyzer (Diesse Diagnostica Senese S.p.A., Monteriggioni, Siena, Italy). CRP was examined using immunofluorescence (UNICELL-S, Shenzhen, Guangdong Province, China).

### 2.9. Statistical Analysis

Descriptive data are reported as the mean ± SD or median (IQR) if they had a normal or non-normal distribution, respectively. The sample size was calculated using the G-Power program, version 3.10 (Düsseldorf, Germany). The partial eta-squared (η^2^) was set at a medium level (0.06) [26]. Assuming a moderate effect size of 0.25, power of 90%, and a dropout rate of 20%, we planned to enroll 52 participants. The parametric distribution was assessed using the Shapiro–Wilk test. Nominal data were summarized as frequencies and percentages. Categorical variables were compared using the chi-squared test or Fisher’s exact test, as appropriate. Continuous variables were compared by independent *t*-test or Mann–Whitney U test, as appropriate. Data were analyzed using the Statistical Package for Social Sciences Version 22.0 (IBM Inc., Armonk, NY, USA). The level of significance was taken at 0.05 or 5%.

## 3. Results

### 3.1. Participant Characteristics

Sixty-six COVID-19 patients were identified as eligible for participation. Fourteen participants were excluded, as shown in Figure 2. Most participants were classified as having the Delta variant (one participant had the Omicron variant). The mean age of all participants was 45 ± 14 years, and 46% were men. Most participants (87%) were vaccinated against COVID-19. The COVID-19 severity was similar between the groups (Table 1 and Table 2). Most participants (n = 49, 94%) were categorized as non-severe cases, which ranged from mild to moderate levels of COVID-19. The proportions of disease severity were as follows: mild (n = 26; 50%), moderate (n = 23; 44%), and severe (n = 3; 6%). Most participants had comorbidities (n = 41, 79%). At the first bedside physiotherapy visit, three Ex-G1 and twelve Ex-G2 patients were admitted to the intensive care unit (ICU). None of the baseline characteristics were significantly different between the groups (all *p* > 0.05) (Table 1 and Table 2). All hematologic parameters were in the normal range, except for the inflammatory markers of ESR and CRP (Table 2).

### 3.2. Primary Outcomes

There were no significant differences in all outcome measures between the intervention groups (*p* > 0.05) (Table 3). The effect sizes for all primary outcomes ranged from small to medium: 0.140 for the survival rate, 0.217 for the length of hospital stay, 0.140 for ICU referrals, and 0.200 for complications. The survival rate of all COVID-19 patients was 98% in this study. The survival rate of COVID-19 patients with mild and moderate conditions was equal (100%) between groups. There was only one Ex-G2 participant referred to the ICU after enrollment in the study due to influenza. There were two Ex-G1 and four Ex-G2 participants with complications after receiving PTPs (Table 3).

### 3.3. Secondary Outcomes

Five Ex-G1 and eleven Ex-G2 COVID-19 patients had minor adverse events during and after the PTPs (Table 3). Among eight Ex-G2 participants, there were 14 sessions (9%) of the bedside PTPs that exhibited a SpO_2_ drop > 3% from baseline (Table 3). None of the participants had serious adverse events during and immediately after the PTPs. There was one Ex-G2 participant who died of cardiac arrest on the day (4 a.m.) after the first bedside PTP (breathing exercise and chest trunk mobilization exercise). This patient had underlying conditions, including obesity (BMI = 57.41), sleep apnea, and pleural effusions. No physiotherapists tested positive for COVID-19 as a consequence of administering bedside PTPs to patients. The number of physiotherapy sessions was significantly different between the groups (*p* < 0.001).

## 4. Discussion

To our knowledge, this is among the first studies to investigate the effects of varying frequencies of administration of bedside physiotherapy programs (PTPs) in hospitalized COVID-19 patients. In addition, there is a lack of study reports on the safety of bedside PTPs in the acute phase of COVID-19, mostly identified as a mild to moderate degree of severity. The main findings indicated no differences between the groups regarding the survival rate, LoH, referrals to the ICU, and in-hospital complications. Overall, there was a high survival rate among the patients, no deaths among mild to moderate COVID-19 patients, a limited number of COVID-19 patients who were referred to the ICU after receiving the PTPs, as well as a low rate of complications observed after receiving the PTPs. Importantly, no serious adverse events occurred during or immediately after each PTP session. In addition, no physiotherapists tested positive for COVID-19 during the two months of the in-hospital data collection period.

To date, few studies on the effects of PTPs have been reported regarding the acute phase of COVID-19 [11,12,14]. In the literature, most previous studies have investigated physical rehabilitation in severe cases of COVID-19 or in patients who were referred to the ICU [11,14] or after recovery from critical illnesses [20,21]. Moreover, a study assessed patients who were negative for SARS-CoV-2 by laboratory diagnostic tests [21] or with long-term hospital stays. This approach was considered safer and posed a lower risk of disease transmission compared to our study, which was conducted with active COVID-19 cases. Interestingly, previous studies have observed that PTPs are safe and feasible in the ICU setting or the post-recovery period [14,20,21]. PTPs also improved patients’ motor and respiratory function, along with functional activity, particularly in post-critical illness patients [14,20,21]. However, these patients were older and had more severe conditions than those in the present study [11,12,14].

The current study suggests that PTPs may provide positive effects on physical function, being similar to a previous study that examined a one-week telerehabilitation program in mild to moderate COVID-19 patients who were confined at home [27]. To support this, there was only one patient who was referred to the ICU, a few complications were found, and none of the patients with mild to moderate conditions died during our intervention in the present study. In addition, our study showed minor complications (9.6%: four bacterial infections, one influenza), compared to an earlier study that reported approximately 39% of complications in patients aged between 19 and 49 years [28]. Moreover, there were no patients who developed ARDS in this study, which is in contrast to a previous one, reporting 23% and 4% of ARDS in the pneumonia group and mild to moderate group, respectively [29]. For these reasons, the survival rate of patients with mild to moderate conditions in the current study was very high, contrary to many previous studies [12,14]. Meanwhile, the LoH of a previous study [20] is comparable to that in the present study, which was 9.8 days. Similarly, the median LoH of patients with pneumonia not caused by COVID-19 was 9 days [30]. In contrast, a recent review study found that the median LoH of COVID-19 patients in China and other countries was 14 and 5 days, respectively. However, during the first year of the COVID-19 pandemic, patients showed a higher LoH compared to the current study [31]. The variations in the LoH could be due to differences in health policies in each country, the development of treatments, and the effectiveness of vaccination. In Thailand, the Ministry of Public Health announced that COVID-19 patients must stay in the hospital for at least 10 days as a control measure against SARS-CoV-2 transmission.

The two most used PTPs, namely breathing exercises and progressive mobilization/exercise, are confirmed to be safe for COVID-19 patients. None of the studied patients experienced serious adverse events during or immediately after the PTPs. This might have been due to the comprehensive screening of patients’ clinical records before and after the PTPs [20]. Among the minor adverse events, a drop in SpO_2_ > 3% from baseline was the most common adverse effect found in the Ex-G2 group, because the participants in the Ex-G2 group were encouraged at the bedside to perform light exercise/ambulation. This accounted for 31% of the participants in Ex-G2; however, there were only 9% (14 sessions) in terms of the total sessions of PTPs in Ex-G2. Nevertheless, the physiotherapist should be cautious in implementing a clinically applicable approach both during and after PTPs.

Our findings support a recent systematic review that found pulmonary rehabilitation to be safe and feasible for COVID-19 patients [32]. In addition, it is in line with a previous recommendation that patients with a mild condition may benefit from breathing exercises [33]. A previous study has also shown a symptomatic improvement after six weeks of an online breathing program for patients with post-COVID-19 [34]. In addition, patients with mild symptoms of SARS-CoV-2 were able to perform mild- to moderate-intensity exercise during active COVID-19 [35]. However, an exercise intensity that is higher than the lactate threshold might not be appropriate according to the production of respiratory droplets [36]. Moreover, previous studies have demonstrated that exercise at a moderate intensity could downregulate inflammatory cytokines and stimulate the release of anti-inflammatory substances [18,19]. Likewise, two weeks of moderate aerobic exercise in COVID-19 patients has recently shown an improvement in immune function [17]. A recent study proposed that exercise promotes myokine production, which would alleviate SARS-CoV-2 virulence [37]. In addition to the physical aspect, the psychological impact may also improve after bedside PTPs, as supported by a previous study [36]. The PTPs in the mentioned study were similar to those in our study, except for the inclusion of craft activities in the earlier study [36]. Therefore, a positive immune system response, the release of anti-SARS-CoV-2 substances, and psychological improvements after PTPs may have contributed to their beneficial effects on most of the primary outcome measures. Notably, we provided the BreatheMax, a breathing biofeedback device, to Ex-G2 patients. From our observations, it can increase SpO_2_ within a few minutes, similarly to a conventional breathing exercise. Thus, this study supports a previous recommendation that PEP or OIS devices can be applied to COVID-19 patients without complications [9]. Interestingly, one to two instances of bedside PTPs provided equal benefits compared to the daily PTP group, which differs from a previous study [23]. This may have been due to the patients in both groups being encouraged to participate in PTPs via a ward phone, private mobile phone application, and closed private groups. Therefore, implementing daily bedside PTPs may be unnecessary for COVID-19 patients with mild to moderate conditions during the acute phase (Figure 1). Telemedicine is an alternative platform in the case of restrictions on bedside PTPs. In addition, this approach allows us to conserve medical equipment during each bedside visit and reduces the risk of COVID-19 transmission from patients. Importantly, our study confirmed that bedside PTPs are safe for physiotherapists, because there were no reports of COVID-19 infection during the two months of prospective data collection among these physiotherapists. Indeed, the physiotherapists in our study strictly followed the airborne, aerosol, and manual contact precautions when providing the bedside PTPs to reduce the possibility of SARS-CoV-2 infection. Based on our findings and similar studies presented in previous publications [38], we provide guidance for physiotherapists in managing COVID-19 patients during both the acute and recovery phases, as illustrated in Figure 1.

There were several limitations in the present study. First, the lack of significant effects may be attributable to low event rates, which could have limited the statistical power to detect meaningful differences. Second, the study did not include a control group, restricting the ability to compare the outcomes with those of conventional treatment. Additionally, randomization procedures and the blinding of participants and assessors were not implemented. These limitations may have influenced the interpretation of the findings and reduced the generalizability of the results. Third, an analysis of objective outcome measures and the psychological status at baseline could not be performed due to unstable signs and symptoms and concerns about COVID-19 transmission, which is reflected in most previous case reports, where researchers were unable to perform these tests [11,12]. In addition, blood biomarkers and chest radiographs could not be obtained before discharge due to limitations imposed by hospital rules. These variables may have been useful to explain the physiological changes after the PTPs. Fourth, according to the quarantine rules, patients could not be discharged from the hospital, even if there were no complications and they had almost fully recovered from COVID-19 infection. Consequently, the LoH among the groups showed no differences. Future studies are highly recommended to confirm the findings of the present study with a conventional treatment control group and in the absence of a mandatory quarantine.

## 5. Conclusions

The different bedside PTP frequencies in COVID-19 patients with primarily mild to moderate conditions resulted in no differences in the survival rate, LoH, referrals to the ICU, and in-hospital complications. PTPs are safe for COVID-19 patients and physiotherapists. The prudent assessment and monitoring of physiological parameters during PTPs are necessary to prevent unpredictable adverse events.

## Figures and Tables

**Figure 1 ijerph-22-00931-f001:**
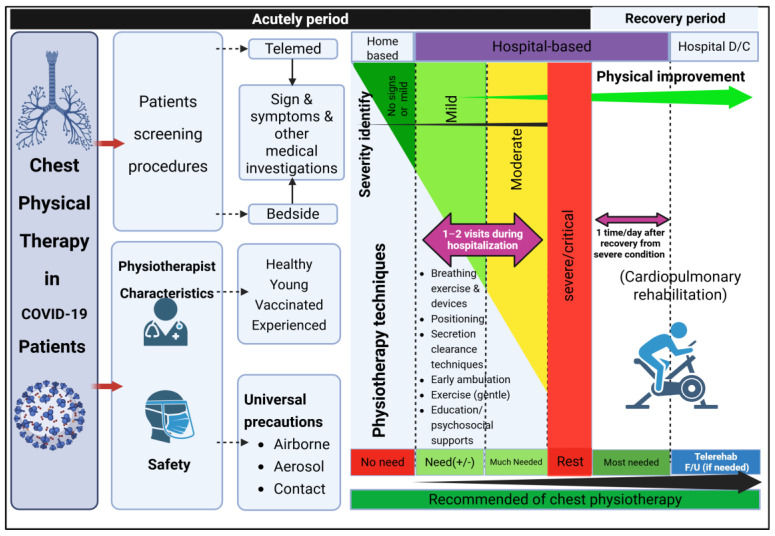
Physiotherapy programs, procedures, and the need for chest physiotherapy in different phases of COVID-19 infection.

**Figure 2 ijerph-22-00931-f002:**
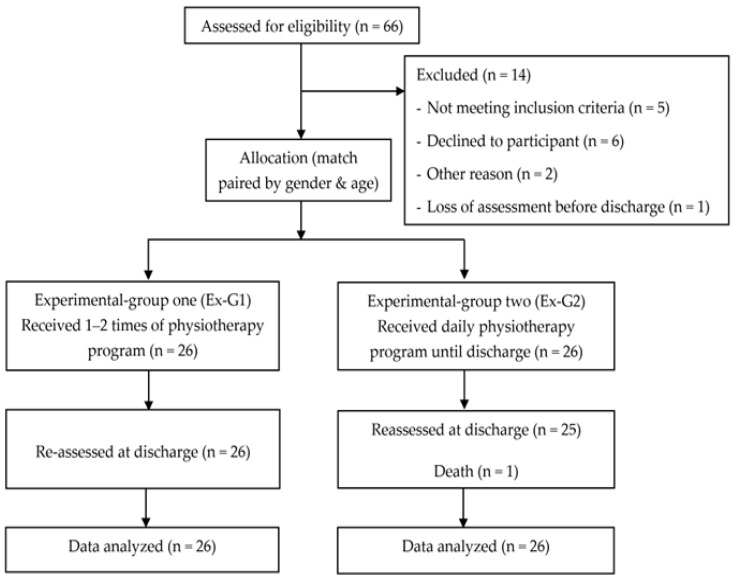
Flow diagram for participants throughout this study.

**Table 1 ijerph-22-00931-t001:** Comparison of baseline characteristics between groups.

	Ex-G1 (n = 26)	Ex-G2 (n = 26)	*p*-Value
Men	12 (46%)	12 (46%)	1.000
Age, mean (SD), years	43.50 (13.34)	45.92 (15.24)	0.545
Age > 60 year	4 (16%)	5 (19%)	0.714
Education level			
Higher than bachelor degree	4 (16%)	1 (4%)	0.168
Bachelor degree	9 (35%)	7 (27%)	
Secondary school	4 (16%)	10 (38%)	
Primary school	7 (27%)	8 (31%)	
No formal education	2 (8%)	0	
Oxygen device used			
Nasal cannula	1 (4%)	3 (12%)	0.350
High-flow nasal cannula	0	1 (4%)	
Mechanical ventilator	0	0	
Comorbidities			
No	4 (15%)	7 (27%)	0.308
Yes ^a^	22 (85%)	19 (73%)	
BW, median (IQR), kg	69.50 (57.00, 89.25)	68.00 (57.00, 85.00)	0.905
BMI, median (IQR), kg/m^2^	25.06 (22.83, 32.34)	26.88 (23.18, 30.99)	0.504
Classification of BMI ^b^			
Obese	13 (50%)	16 (62%)	0.606
Overweight	6 (23%)	5 (19%)	
Normal	7 (27%)	5 (19%)	
COVID-19 disease severity			
Mild	15 (58%)	11 (42%)	0.538
Moderate	10 (38%)	13 (50%)	
Severe	1 (4%)	2 (8%)	
Duration from admission to initiation PT, mean (SD), days	1.62 (0.80)	1.69 (1.09)	0.773
Antiviral medication use			
Favipiravir	24 (92%)	22 (100%)	1.000
Mixed antiviral drug	13 (50%)	10 (40%)	0.402
Corticosteroid drug	3 (12%)	8 (32%)	0.090

Data expressed as n (%) unless otherwise stated. ^a^ Comorbidities included hypertension, diabetes mellitus, cerebrovascular events, thyroid issues, allergies, glucose-6-phosphate dehydrogenase deficiency, chronic hepatitis B infection, thalassemia, systemic lupus erythematosus, enlarged prostate, gout, old tuberculosis infection, migraine, and chronic kidney disease. ^b^ Obese = BMI ≥ 25; overweight = 23 ≤ BMI < 25; and normal = 18.5 ≤ BMI < 23. BMI, body mass index; BW, body weight; COVID-19, coronavirus disease 2019; IQR, interquartile range; PT, physiotherapy; SD, standard deviation.

**Table 2 ijerph-22-00931-t002:** Comparison of baseline blood biomarkers between groups.

	Ex-G1 (n = 26)	Ex-G2 (n = 25 ^a^)	*p*-Value
ORF1ab gene, Ct	20.81 (17.76, 23.64)	20.56 (18.59, 24.10)	0.859
E gene, Ct	21.70 (18.09, 24.09)	21.07 (19.03, 23.93)	0.861
Complete blood count			
WBC, mean (SD), ×10^3^/uL	6.90 (1.74)	7.01 (2.44)	0.856
RBC, mean (SD), 10^6^/uL	5.07 (0.61)	4.83 (0.64)	0.167
HGB, mean (SD), g/dL	13.27 (1.56)	13.4 (1.45)	0.661
HCT, mean (SD), %	40.62 (4.44)	40.84 (4.01)	0.850
RDW, %	13.90 (13.00, 14.53)	13.50 (12.60, 13.90)	0.086
Platelet, ×10^3^/uL	254.00 (215.50, 293.75)	254.00 (207.50, 332.00)	0.706
Neutrophil, mean (SD), %	60.77 (13.39)	63.56 (12.94)	0.453
Lymphocyte, mean (SD), %	28.46 (12.34)	26.84 (12.83)	0.647
Monocyte, %	7.50 (6.00, 10.00)	6.00 (5.00, 9.50)	0.209
Eosinophil, %	1.00 (0.00, 3.00)	1.00 (0.00, 2.00)	0.486
Basophil, %	0.00 (0.00, 1.00)	0.00 (0.00, 1.00)	0.925
Kidney function test			
BUN, mg/dL	12.50 (9.00, 15.00)	11.00 (8.00, 17.00)	0.769
Creatinine, mg/dL	0.74 (0.64, 0.88)	0.71 (0.58, 1.04)	0.850
eGFR, mL/min/1.73 m^2^	106.01 (87.45, 115.83)	107.06 (71.72, 122.14)	0.880
eGFR stage, n (%)			
Stage I	19 (73%)	17 (68%)	0.086
Stage II	7 (27%)	4 (16%)	
Stage III	0	4 (16%)	
Liver function test			
Total protein, mean (SD), g/dL	7.74 (0.75)	7.63 (0.58)	0.655
Total bilirubin, mean (SD), mg/dL	0.49 (0.18)	0.52 (0.17)	0.855
Alkaline phosphate, U/L	69.00 (55.00, 88.00)	67.00 (52.00, 78.50)	0.516
AST/SGOT, U/L	26.00 (22.50, 36.50)	26.00 (23.00, 30.00)	0.437
ALT/SGPT, U/L	24.00 (17.00, 47.50)	22.00 (14.00, 39.00)	0.166
LDH, U/L	189.30 (159.50, 207.15)	176.30 (163.55, 242.85)	0.777
Inflammation biomarkers			
ESR, mean (SD), mm/h	26.92 (13.06)	31.12 (19.00)	0.297
CRP, mg/dL	8.26 (3.56, 17.94)	13.89 (5.80, 25.66)	0.122

Data expressed as median (IQR) unless otherwise stated. ^a^ n = 25 as a laboratory test in one participant was not required by the clinician. ALT/SGPT, alanine aminotransferase/serum glutamic–pyruvic transaminase; AST/SGOT, aspartate aminotransferase/serum glutamic–oxaloacetic transaminase; BUN, blood urine nitrogen; CRP, C-reactive protein; Ct, threshold cycle; E gene, envelope gene; eGFR, estimated glomerular filtration rate; ESR, erythrocyte sedimentation rate; HCT, hematocrit; HGB, hemoglobin; LDH, lactate dehydrogenase; ORF1ab gene, open reading frame 1ab gene; RBC, red blood cell; RDW, red blood cell distribution width; WBC, white blood cell.

**Table 3 ijerph-22-00931-t003:** Outcome measurement comparison between groups.

	Ex-G1 (n = 26)	Ex-G2 (n = 26)	*p*-Value
Survival	26 (100%)	25 (96%)	1.000
Death	0	1 (4%)	1.000
LoH, median (IQR), days	10.00 (9.00, 11.80)	10.00 (10.00, 12.00)	0.117
Patients referred to ICU	0	1 (4%)	0.313
Complications			
Influenza	0	1 (4%)	0.555
Bacterial infections	2 (8%)	2 (8%)	
Cardiac arrest	0	1 (4%)	
None	24 (92%)	22 (84%)	
Minor adverse event			
Drop in SpO_2_ > 3% from baseline	0	8 (31%)	0.018
Dizziness	1 (4%)	1 (4%)	
Nausea and vomiting	2 (8%)	0	
Dyspnea ^a^	2 (8%)	1 (4%)	
Dyspnea and drop in SpO_2_ > 3% from baseline ^a^	0	1 (4%)	
None	21 (80%)	15 (57%)	
Average of PT bedside, median (IQR)	2.00 (1.00, 2.00)	6.00 (5.00, 7.00)	<0.001
Number of patients receiving each PT program			
Breathing exercise, cough/huff training, active chest trunk mobilization, positioning, active exercise of UE and LE	26 (100%)	26 (100%)	1.000
Positive expiratory pressure devices	0	26 (100%)	<0.001
Out-of-bed exercise	0	26 (100%)	<0.001

Data expressed as n (%) unless otherwise stated. ^a^ Dyspnea caused by continued cough during breathing exercise. ICU, intensive care unit; IQR, interquartile range; LE, lower extremity; LoH, length of hospital stay; PT, physiotherapy; UE, upper extremity.

## Data Availability

The authors will make all relevant data available upon reasonable request.

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
