# Peer review of "Effects of Different Bedside Physiotherapy Frequencies in Hospitalized COVID-19 Patients, Focusing on Mild to Moderate Cases"

_ijerph, 2025, doi:10.3390/ijerph22060931_

Round 1

Reviewer 1 Report

Comments and Suggestions for Authors

The manuscript (ijerph-3611171) explores quasi-experimental study assessing the effect of various frequencies of bedside PTPs on clinical results among hospitalized COVID-19 patients with mainly mild to moderate severity. Although, the study is written well and objectives and methodology were clear, but there are several aspects that needs to be clarified and improved for possible publication. The authors should look into following comments:

  1. The author should include negative control group indicating non treated group, as its inclusion will strengthen the conclusions related to the efficacy of PTPs.
  2. The criteria employed for classification of disease severity needs to be further clarified, mainly because including patients with oxygen necessity and use of corticosterioid may disort the differences between moderate and severe groups.
  3. Author should include details with regards to assumptions regarding anticipated effect size and focus of outcome measure.
  4. The outcomes discussed in the study, such as survivial, ICU transfer, LoH may not evidently determine functional or quality-of-life advantages of various physiotherapy regimens. Authors should consider whether secondary functional or patient-related outcomes could improve interpretation.
  5. Authors reported no serious adverse events, however, there was no death due to cardiac arrest in PTP group. Although, the incident was briefly included, but additional details should included with regards to baseline characteristics of patients, timing of events, and if death was probably due to PTPs.
  6. There are minor grammatical and typographical errors are there throughout the manuscript. It should be corrected via thorough proofreading or correction by taking help form native English speaker.

Author Response

Reviewer I

Comments and Suggestions for Authors

The manuscript (ijerph-3611171) explores quasi-experimental study assessing the effect of various frequencies of bedside PTPs on clinical results among hospitalized COVID-19 patients with mainly mild to moderate severity. Although, the study is written well and objectives and methodology were clear, but there are several aspects that needs to be clarified and improved for possible publication. The authors should look into following comments:

Comments 1: The author should include a negative control group indicating non treated group, as its inclusion will strengthen the conclusions related to the efficacy of PTPs.

Response 1: We appreciate the reviewer’s insightful comment regarding the inclusion of a negative control group. We acknowledge that our study did not include a non-treated control group, as both cohorts received physiotherapy interventions. As such, we were unable to compare outcomes against a group that received no physiotherapy. Furthermore, our findings revealed no significant differences between groups receiving different frequencies of PTPs. To address this limitation and enhance clarity, we have revised the manuscript to include the following statement: “Implementing daily bedside PTPs may be unnecessary for COVID-19 patients with mild to moderate conditions during the acute phase” (page 14, lines 345–347). We recognize that future studies incorporating a true control group would be valuable in further evaluating the efficacy of PTPs (page 14, line 374).   

Comments 2: The criteria employed for classification of disease severity needs to be further clarified, mainly because including patients with oxygen necessity and use of corticosterioid may disort the differences between moderate and severe groups.

Response 2: Thank you for the suggestion. To minimize potential confusion for readers, we have removed the statement: “A patient who has been administered a combination of antiviral drugs and/or received corticosteroid treatment was classified as a moderate to severe case.” Furthermore, we have added additional details regarding the criteria for COVID-19 classification on page 3, lines 123–128.

Comments 3: Author should include details with regards to assumptions regarding anticipated effect size and focus of outcome measure.

Response 3: We have included the effect sizes for the primary outcomes, as presented on page 11, lines 240–242. 

Comments 4: The outcomes discussed in the study, such as survival, ICU transfer, LoH may not evidently determine functional or quality-of-life advantages of various physiotherapy regimens. Authors should consider whether secondary functional or patient-related outcomes could improve interpretation.

Response 4: Thank you for the comment. However, our study did not assess functional outcomes based on awareness of COVID-19 transmission. We have acknowledged this as a limitation and suggested it as an area for future research on page 14, lines 364–367.

Comments 5: Authors reported no serious adverse events, however, there was no death due to cardiac arrest in PTP group. Although, the incident was briefly included, but additional details should included with regards to baseline characteristics of patients, timing of events, and if death was probably due to PTPs.

Response 5: Thank you for your thoughtful comment. In response, we have provided additional details regarding the patient who experienced cardiac arrest, including the patient's baseline characteristics, underlying conditions, and the physiotherapy interventions received. These details have been added to the revised manuscript (page 12, lines 256–259). Importantly, the cardiac arrest occurred the following morning at approximately 4 a.m., not immediately after the physiotherapy session. Based on the timing and clinical context, we believe that this event was unrelated to the PTP intervention.

Comments 6: There are minor grammatical and typographical errors are there throughout the manuscript. It should be corrected via thorough proofreading or correction by taking help from native English speaker.

Response 6: The manuscript has been reviewed for grammatical and typographical accuracy by a native English speaker. However, if further refinement is required, we are willing to undertake additional proofreading.

Reviewer 2 Report

Comments and Suggestions for Authors

Respected Authors,

Below are the comments / suggestions of mine.

  1. Line 24 - Do you have any proof to confirm that no studies were ever conducted to find the impact of PTP in COVID-19?
  2. Line 52 & 59 - What was the criteria for a person to be considered as an expert?
  3. Line 68 - You have mentioned 'Some of the recommendations'. But provided only one reference.
  4. Line 72 - Same as above.
  5. Line 122 - Intervention: Whats the difference between Ex-G1 and Ex-G2 If participants in both the groups receive same Physiotherapy treatment?
  6. Line 126 - When a Physiotherapy treatment was taught to the participants, it is expected that they perform those exercises regularly / frequently. How this is different from Ex-G2?
  7. Line 136 - What is the specific reason behind teaching the participants of Ex-G1 only one-time?
  8. Why there is no control group in this study (Line 323)?
  9. Line 181 - Why did the authors choose these specific determinants to determine the sample size?
  10. Line 200 - 3 patients were admitted in the ICU. Where they included in this study or excluded?
  11. If those 3 were included, then the line 215 is contradicting. Kindly verify.
  12. Several baseline characteristics were observed (Table 1 and 2). But they were not properly analyzed or discussed in Table 3 or in the discussion part. E.g.: Why patients education levels were observed? How does it play a role in your study?
  13. Line 234 - Have the authors conducted a Scoping or Systematic Review before mentioning this statement?
  14. This study talks about the frequency of PTP. What were the PTP frequency of Ex-G1 and Ex-G2? Generally, it is assumed that more frequency leads to more efficient outcome. E.g.: When a Doctor prescribes a medication, we dont take them just 1 or 2 days. But, for a whole sitting / duration. So, how the authors expect that 1 or 2 days of PTP can provide a better outcome than a continuous sessions of PTP? Do you have any explanations?

Regards.

Author Response

Reviewer II

Comments 1: Line 24 - Do you have any proof to confirm that no studies were ever conducted to find the impact of PTP in COVID-19?

Response 1: Thank you for your valuable comment. Upon further literature review, we identified a limited number of studies that specifically examined the impact of different frequencies of physical therapy in patients with COVID-19. To reflect this more accurately, we have revised the wording in the manuscript and replaced the phrase “no studies” with a more appropriate alternative to enhance clarity (page 1, lines 29–30).

Comments 2: Line 52 & 59 - What was the criteria for a person to be considered as an expert?

Response 2: Thank you for your comment. Based on our literature review (see references below), the term “expert” refers to researchers, physicians (e.g., pulmonologists), specialists in respiratory or cardiorespiratory care, and physiotherapists affiliated with national physical therapy associations in countries such as the United States, Canada, and Japan. To improve clarity and precision, we have revised the manuscript accordingly (page 2, lines 57–58).

Reference:

No.8 Felten-Barentsz, K.M.; van Oorsouw, R.; Klooster, E.; Koenders, N.; Driehuis, F.; Hulzebos, E.H.J., et al., Recommendations for hospital-based physical therapists managing patients with COVID-19. Phys Ther., 2020. 100(9): p. 1444-1457.

No.9 Thomas, P.; Baldwin, C.; Beach, L.; Bissett, B.; Boden, I.; Cruz, S.M., et al., Physiotherapy management for COVID-19 in the acute hospital setting and beyond: an update to clinical practice recommendations. J Physiother., 2022. 68(1): p. 8-25.

No.10 Vitacca, M.; Carone, M.; Clini, E.M.; Paneroni, M.; Lazzeri, M.; Lanza, A., et al., Joint statement on the role of respiratory rehabilitation in the COVID-19 crisis: the Italian position paper. Respiration, 2020. 99(6): p. 493-499.

Comments 3: Line 68 - You have mentioned 'Some of the recommendations'. But provided only one reference.

Response 3: We apologize for the error in our English writing. We intended to refer to recommendations from a narrative review study, and have accordingly revised the wording on page 2, lines 75–77.

Comments 4: Line 72 - Same as above.

Response 4: We have already changed the wording from 'some study' to 'One study found that…' on page 2, line 80.

Comments 5: Line 122 - Intervention: What's the difference between Ex-G1 and Ex-G2 If participants in both the groups receive same Physiotherapy treatment?

Response 5: Thank you for your comment. As noted in the manuscript (page 2, lines 85–87), this study aimed to compare the effects of different frequencies of PTPs. While both groups received the same type of physiotherapy treatment, participants in the Ex-G2 group additionally used some breathing devices. The primary difference between groups was the frequency of sessions: participants in the Ex-G1 group received PTPs only 1–2 times during their hospital stay, whereas those in the Ex-G2 group received daily PTPs until discharge. This information is provided in detail on page 3, lines 135–142, and page 4, lines 143-147.

Comments 6: Line 126 - When a Physiotherapy treatment was taught to the participants, it is expected that they perform those exercises regularly / frequently. How this is different from Ex-G2?

Response 6: Patients in both groups received physiotherapy programs (PTPs) as soon as possible after hospital admission, as stated on page 3, lines 135–136 of the manuscript. The primary difference between groups was the frequency of bedside visits. Patients in Experimental Group Two (Ex-G2) received daily PTPs until discharge, whereas those in Experimental Group One (Ex-G1) received PTPs only once during admission. For patients who were highly confused and unable to understand the instructions, physiotherapists repeated bedside teaching, but no more than twice during the hospital stay. Details of the PTPs for each group are provided on page 3, lines 135–142, and page 4, lines 143-147.

Comments 7: Line 136 - What is the specific reason behind teaching the participants of Ex-G1 only one-time?

Response 7: We apologize for the lack of clarity regarding the frequency of PTPs in Ex-G1. Patients in Ex-G1 received a single bedside physiotherapy session, as daily visits were difficult to conduct. This limitation arose from the need to minimize the risk of COVID-19 transmission and to reduce the cost of bedside care, particularly given the limited availability of essential equipment such as N95 masks and personal protective equipment (PPE) in Thailand.

Furthermore, established guidelines for physiotherapy during the acute phase of COVID-19 recommend that physiotherapy staff should not routinely enter isolation rooms (see references below). In addition, certain physiotherapy interventions—such as percussion, vibration, and assisted coughing—are classified as aerosol-generating procedures (AGPs), which may further increase the risk of viral transmission. (Some of the above information was mentioned in page 14, lines 348-349)

Therefore, the reduced frequency of bedside physiotherapy during the acute phase may have posed a challenge for patients. These considerations led us to compare different frequencies of PTPs between the study groups.

Reference

1.Thomas P, Baldwin C, Bissett B, Boden I, Gosselink R, Granger CL, et al. Physiotherapy management for COVID-19 in the acute hospital setting: clinical practice recommendations. J Physiother. 2020;66(2):73-82.

  1. Thomas P, Baldwin C, Beach L, Bissett B, Boden I, Cruz SM, et al. Physiotherapy management for COVID-19 in the acute hospital setting and beyond: an update to clinical practice recommendations. J Physiother. 2022;68(1):8-25.

Comments 8: Why there is no control group in this study (Line 323)?

Response 8: Thank you for the important question. We did not include a control group in this study for several reasons:

  1. Certain physical therapy treatments—such as breathing exercises or secretion removal techniques in patients with excessive phlegm—are considered standard care in conditions like bacterial pneumonia and COPD. Withholding these interventions would raise ethical concerns, particularly in patients where such treatments are clinically indicated, and this is the most important reason.
  2. We faced resource limitations, including budget constraints and limited staffing, which made it impractical to add an additional study group.
  3. There were time constraints during the study period that further limited our ability to expand the study design to include a control arm. 

This information has already been included in the manuscript (page 3, lines 102-104).

Comments 9: Line 181 - Why did the authors choose these specific determinants to determine the sample size?

Response 9: We determined the sample size using the G*Power program, applying a medium effect size based on partial eta squared (η² = 0.06), which is appropriate for a two-parallel arm design. The corresponding reference has been included in the manuscript (reference number 26). While 80% power is a standard and widely accepted threshold in clinical study design, we opted for a higher power level of 90% in our study. This decision was made to reduce the risk of Type II error (false negatives), minimize the chance of overlooking a true effect, and enhance the clinical relevance and reliability of the study findings.

Reference No. 26

Lakens D. Calculating and reporting effect sizes to facilitate cumulative science: a practical primer for t-tests and ANOVAs. Frontiers in Psychology. 2013;Volume 4 - 2013.

Comments 10: Line 200 - 3 patients were admitted in the ICU. Where they included in this study or excluded?

Response 10: We included COVID-19 patients who were admitted to the ICU, as most had comorbidities and obesity—both of which are significant risk factors for developing acute respiratory distress syndrome (ARDS). Although these patients were not initially classified as severe cases, physicians admitted them to the ICU for close monitoring of their clinical signs and symptoms due to their elevated risk.  

Comments 11: If those 3 were included, then the line 215 is contradicting. Kindly verify.

Response 11: All of the above information is accurate. Three patients were admitted to the ICU on the first day of hospitalization. However, as noted in line 215 (now updated to lines 244–245), one additional patient was subsequently transferred to the ICU due to complications arising from influenza.

Comments 12: Several baseline characteristics were observed (Table 1 and 2). But they were not properly analyzed or discussed in Table 3 or in the discussion part. E.g.: Why patients education levels were observed? How does it play a role in your study?

Response 12: Thank you for your concern. However, we did not provide a detailed discussion of most baseline characteristics, as our findings showed no significant differences between groups. However, education level was included in the data collection because, based on existing knowledge, a lower level of education might present challenges in learning and adhering to the intervention. This might affect the outcome measurement.

Comments 13: Line 234 - Have the authors conducted a Scoping or Systematic Review before mentioning this statement?

Response 13: Thank you for your comment. Yes, we have conducted a systematic review on COVID-19 rehabilitation. However, to date, few studies have specifically compared the effects of different frequencies of physiotherapy programs. We have revised the wording to better reflect this point (page 12, line 264). Most existing studies have focused on the acute or subacute phases in severe cases, or on the outpatient phase, likely to minimize the risk of COVID-19 transmission. This gap in the literature has already been highlighted in the Introduction section (page 2, lines 70–74).

Comments 14: This study talks about the frequency of PTP. What were the PTP frequency of Ex-G1 and Ex-G2? Generally, it is assumed that more frequency leads to more efficient outcome. E.g.: When a Doctor prescribes a medication, we dont take them just 1 or 2 days. But, for a whole sitting / duration. So, how the authors expect that 1 or 2 days of PTP can provide a better outcome than a continuous sessions of PTP? Do you have any explanations?

Response 14:  Thank you for your comment. In general, we agree with the reviewer’s observation that more frequent treatment may provide greater benefits compared to less frequent treatment. This approach could be particularly beneficial for patients recovering from severe COVID-19. However, there is limited knowledge regarding its effects in patients with mild to moderate COVID-19, which motivated us to conduct the present study.

Our findings showed no significant difference between groups. As explained on page 14, lines 341–349, this outcome may be attributed to the fact that patients in our study were encouraged to perform PTPs through various channels, including ward phones, private mobile applications, and closed private groups. Notably, limiting bedside PTP visits to one or two sessions may benefit physiotherapists by reducing their risk of COVID-19 transmission and conserving medical equipment.

Reviewer 3 Report

Comments and Suggestions for Authors

Thank you for the opportunity to review this timely and relevant manuscript. The study addresses an important clinical question regarding bedside physiotherapy frequency in hospitalized COVID-19 patients with mild to moderate disease. The manuscript is clearly written, and the methods are well-detailed. I appreciate the authors’ effort and would like to offer two points for consideration that could further strengthen the manuscript:

1. Safety Outcome Interpretation

The fact that 31% of patients in the daily PTP group experienced a >3% drop in SpOâ‚‚ during sessions is notable. While this was classified as a minor adverse event and no serious harm occurred, the relatively high rate in a mild/moderate population deserves a more cautious interpretation. A short discussion on this point—especially regarding safety monitoring and clinical applicability—would enhance the balance of the findings.

Author Response

Reviewer III

Thank you for the opportunity to review this timely and relevant manuscript. The study addresses an important clinical question regarding bedside physiotherapy frequency in hospitalized COVID-19 patients with mild to moderate disease. The manuscript is clearly written, and the methods are well-detailed. I appreciate the authors’ effort and would like to offer two points for consideration that could further strengthen the manuscript:

Comment: Safety Outcome Interpretation

The fact that 31% of patients in the daily PTP group experienced a >3% drop in SpOâ‚‚ during sessions is notable. While this was classified as a minor adverse event and no serious harm occurred, the relatively high rate in a mild/moderate population deserves a more cautious interpretation. A short discussion on this point—especially regarding safety monitoring and clinical applicability—would enhance the balance of the findings.

Response:  We appreciate this valuable suggestion. Although this minor adverse event was observed in 31% of participants (8 out of 26 patients), it occurred in only 14 sessions, representing 9% of the total physiotherapy sessions (page 12, line 254). We agree with the reviewer that the clinical implications of this finding should be addressed. Therefore, we have included a more detailed discussion of the minor adverse event on page 13, lines 316–318.

Reviewer 4 Report

Comments and Suggestions for Authors

This manuscript addresses an important and underexplored area of COVID-19 rehabilitation by comparing outcomes between different frequencies of bedside physiotherapy in hospitalized patients with primarily mild to moderate disease. The study is timely, well-conceived, and executed with appropriate methodological rigor. I will provide a section-by-section report highlighting strengths and area of improvement.

Title and Abstract: The title is informative and the abstract is well-structured, with key findings adequately summarized. The abstract could benefit from clearer statistical detail to support the claim of "no significant difference" between groups.

Introduction: The background summarizes well the current knowledge about physiotherapy in COVID-19. The rationale is we--articulated and the hypothesis clearly stated. However, a clearer distinction between existing evidence in severe vs. mild/moderate cases would strengthen the rationale.

Materials and methods: The methodology is detailed and inclusion/exclusion criteria are clearly defined. There are some areas of improvement though: a) please clarify whether allocation was randomized or just stratified; b) the absence of a control group receiving no physiotherapy should be acknowledged more directly as a design limitation; and c) the monitoring and fidelity of PTP adherence in the Ex-G1 group (who received fewer sessions) could be more clearly explained.

Results: The presentation of data is clear, with comprehensive tables. Authors could further improve the presentation of results by including effect sizes or confidence intervals to better understand the clinical relevance of non-significant findings. Moreover, authors could consider noting the timing of adverse events in relation to PTP sessions, to enhance the value of the reporting.

Discussion and Conclusions: These sections are well-developed and informative. The results are contextualized with the broader literature and thoroughly discussed, and conclusions are clear. Still, some improvements could be considered, such as giving more emphasis to the limitations of lack of effect (possibly due to low event rates) and of lack of blinding and control.

Minor typos to be corrected.

Author Response

Reviewer IV

This manuscript addresses an important and underexplored area of COVID-19 rehabilitation by comparing outcomes between different frequencies of bedside physiotherapy in hospitalized patients with primarily mild to moderate disease. The study is timely, well-conceived, and executed with appropriate methodological rigor. I will provide a section-by-section report highlighting strengths and area of improvement.

Comment 1: Title and Abstract: The title is informative and the abstract is well-structured, with key findings adequately summarized. The abstract could benefit from clearer statistical detail to support the claim of "no significant difference" between groups.

Response 1: We appreciate the reviewer’s positive feedback on the title and abstract. In response to the suggestion for more clarity regarding statistical findings, we have revised the abstract to include key statistical values (e.g., p-values) to support the statement of “no significant difference” between groups. These additions aim to enhance the transparency and precision of the reported outcomes (page 1, lines 38-40).

Comment 2: Introduction: The background summarizes well the current knowledge about physiotherapy in COVID-19. The rationale is well articulated and the hypothesis clearly stated. However, a clearer distinction between existing evidence in severe vs. mild/moderate cases would strengthen the rationale.

Response 2: We thank the reviewer for the insightful comment. We have revised the introduction to more clearly distinguish the existing evidence regarding physiotherapy in patients with severe versus mild to moderate COVID-19. Specifically, we have added a paragraph highlighting that most existing studies and guidelines primarily focus on physiotherapy management in severe or critically ill patients, while evidence in mild to moderate cases remains limited and inconclusive, page 2, lines 70-77.

Comment 3: Materials and methods: The methodology is detailed and inclusion/exclusion criteria are clearly defined. There are some areas of improvement though: a) please clarify whether allocation was randomized or just stratified; b) the absence of a control group receiving no physiotherapy should be acknowledged more directly as a design limitation; and c) the monitoring and fidelity of PTP adherence in the Ex-G1 group (who received fewer sessions) could be more clearly explained.

Response 3: Thank you for your valuable suggestions regarding the Methods section.

  1. a) We have clarified the allocation procedure, specifying that participants were stratified by age and gender and then sequentially assigned to the two groups based on the order of hospital admission (page 3, lines 101–102).
  2. b) The absence of a control group receiving no physiotherapy has been explicitly acknowledged as a study limitation, along with the underlying ethical considerations and constraints related to time and budget (page 3, lines 102–104).
  3. c) Additional details regarding the monitoring of adherence and the strategies implemented to promote compliance in both groups have been provided to improve clarity (page 4, lines 146–154).

Comment 4: Results: The presentation of data is clear, with comprehensive tables. Authors could further improve the presentation of results by including effect sizes or confidence intervals to better understand the clinical relevance of non-significant findings. Moreover, authors could consider noting the timing of adverse events in relation to PTP sessions to enhance the value of the reporting.

Response 4: We thank the reviewer for this valuable suggestion. We have revised the results section to include effect sizes for all the primary outcomes (page 11, lines 240-242) Additionally, we have specified the timing of the serious adverse event (death) involving one patient to provide a clearer understanding of when it occurred (page 12, lines 256–257). Further relevant details about this patient are also provided on page 12, lines 258–259.

Comment 5: Discussion and Conclusions: These sections are well-developed and informative. The results are contextualized with the broader literature and thoroughly discussed, and conclusions are clear. Still, some improvements could be considered, such as giving more emphasis to the limitations of lack of effect (possibly due to low event rates) and of lack of blinding and control.

Response 5: We again thank the reviewer for their encouraging feedback and constructive suggestions. We now address the possibility that the lack of significant effects may be attributable to low event rates, which could have limited the statistical power to detect meaningful differences (page 14, lines 358-360). Additionally, we have expanded our discussion of the methodological limitations—namely, the absence of randomization procedures, blinding, and a control group—and how these factors may have impacted the interpretation and generalizability of the results (page 14, lines 360–364). 

Round 2

Reviewer 1 Report

Comments and Suggestions for Authors

Manuscript can be accepted for publication in its current form.